# Myeloid Trem2 Dynamically Regulates the Induction and Resolution of Hepatic Ischemia-Reperfusion Injury Inflammation

**DOI:** 10.3390/ijms24076348

**Published:** 2023-03-28

**Authors:** Sheng Han, Xiangdong Li, Nan Xia, Yu Zhang, Wenjie Yu, Jie Li, Chenyu Jiao, Ziyi Wang, Liyong Pu

**Affiliations:** 1Hepatobiliary Center, The First Affiliated Hospital of Nanjing Medical University, Nanjing 210029, China; 2Key Laboratory of Liver Transplantation, Chinese Academy of Medical Sciences, Nanjing 210029, China; 3NHC Key Laboratory of Living Donor Liver Transplantation, Nanjing Medical University, Nanjing 210029, China

**Keywords:** liver ischemia reperfusion, efferocytosis, Trem2, Cox2, PGE2

## Abstract

Trem2, a transmembrane protein that is simultaneously expressed in both bone marrow-derived and embryonic-derived liver-resident macrophages, plays a complex role in liver inflammation. The unique role of myeloid Trem2 in hepatic ischemia-reperfusion (IR) injury is not precisely understood. Our study showed that in the early stage of inflammation induction after IR, Deletion of myeloid Trem2 inhibited the induction of iNOS, MCP-1, and CXCL1/2, alleviated the accumulation of neutrophils and mitochondrial damage, and simultaneously decreased ROS formation. However, when inflammatory monocyte-macrophages gradually evolved into CD11b^hi^Ly6C_low_ pro-resolution macrophages through a phenotypic switch, the story of Trem2 took a turn. Myeloid Trem2 in pro-resolution macrophages promotes phagocytosis of IR-accumulated apoptotic cells by controlling Rac1-related actin polymerization, thereby actively promoting the resolution of inflammation. This effect may be exercised to regulate the Cox2/PGE2 axis by Trem2, alone or synergistically with MerTK/Arg1. Importantly, when myeloid Trem2 was over-expressed, the phenotypic transition of monocytes from a pro-inflammatory to a resolution type was accelerated, whereas knockdown of myeloid Trem2 resulted in delayed upregulation of CX3CR1. Collectively, our findings suggest that myeloid Trem2 is involved in the cascade of IR inflammation in a two-sided capacity, with complex and heterogeneous roles at different stages, not only contributing to our understanding of sterile inflammatory immunity but also to better explore the regulatory strategies and intrinsic requirements of targeting Trem2 in the event of sterile liver injury.

## 1. Introduction

Liver ischemia-reperfusion (IR) injury is an unavoidable pathological reaction during clinical liver transplantation and partial liver resection. Damage-related molecular patterns (DAMPs), including HMGB1 and DNA, drive the development of inflammatory hepatocyte injury through pattern recognition receptors [1,2]. In the context of sterile inflammation, macrophages from different sources may have both pro-inflammatory and anti-inflammatory effects that damage or protect the liver [3,4,5,6]. Taking macrophages in liver IR injury as an example, the induction and resolution of inflammation depend on the common division of labor between monocyte macrophages, resident Kupffer cells, and peritoneal macrophages [7]. Rapid and stable resolution of inflammation is critical for the prognosis of IRI. Macrophage efferocytosis is indispensable for the implementation of an inflammatory resolution program.

Trem2 is a transmembrane protein that can recognize a variety of ligands, including DNA and lipoproteins [8,9], and was previously found to transmit intracellular signals that maintain microglial responses during Alzheimer’s disease. In the liver, Trem2 has been proven to be expressed in resident non-parenchymal cells (NPC, including Kupffer cells and hepatic stellate cells) and recruited neutrophils and monocytes in response to multiple types of liver injury [10,11,12]. Trem2 has been considered an inhibitor of inflammation for a long time [13], and the systemic deficiency of Trem2 responds to n-acetyl-p-aminophenoln-(APAP)-induced exacerbation of liver injury [14]. Moreover, macrophage dynamics and refilling of the Kupffer cell pool in acute liver injury were also impaired by global Trem2 knockout, but the cellular targeting of Trem2 was not clearly distinguished [15]. Correspondingly, some studies have shown that Trem2 can exacerbate the inflammation of microglia with a high-glucose background [16]. Systemic knockout of Trem2 can promote HCC progression, and Trem2 can protect the liver from HCC [10]. However, some studies have shown that antagonizing Trem2 can enhance the immunotherapeutic efficacy in various tumors, including HCC [17]. These conflicting conclusions may be due to confusion regarding the target of Trem2, which makes the specific mechanism elusive.

Since monocytes are recruited to the liver in response to liver IR injury and participate in the complex process of regulating pro-inflammatory and anti-inflammatory activities, we speculate that targeting myeloid Trem2 may play an irreplaceable role in this process. We evaluated the expression of Trem2 in the inflammatory induction and resolution stages of the liver in mice with liver IR injury and explored the specific behavior and corresponding regulatory mechanism of myeloid Trem2 in liver IR injury by constructing Trem2*^FL/FL^*, Trem2*^mKO^*, and Trem2*^creERT^*ROSA26*^TdTomato^* mice.

## 2. Results

### 2.1. The Expression of Trem2 in Monocyte-Derived Macrophages Was in Response to Activated Reperfusion Injury after Hepatic Ischemia

The protein level of Trem2 in the liver responded to significantly activated liver IR (Figure 1A), consistent with the upregulation of transcriptional levels (Figure 1B). Subsequently, hepatocytes and macrophages were isolated at different time points after IR treatment. The expression of Trem2 in hepatocytes and macrophages detected by qPCR and flow cytometry showed that Trem2 was not activated in hepatocytes but was significantly upregulated in macrophages (Figure 1C,D,F). The expression of Trem2 protein in isolated macrophages was significantly activated by the induction of liver inflammation (Figure 1E,F). IHC and immunofluorescence detection of the expression of Trem2 also demonstrated that Trem2 was significantly infiltrated into the IR injury area (Figure 1G). In addition, the IR-induced expression of ALT and AST was significantly correlated with the abundance of Trem2 in the liver (Figure 1H,I). The expression of Trem2 also reflected the degree of tissue damage, as marked by Suzuki’s scores (Figure 1J).

### 2.2. Deficiency of Myeloid Trem2 Alleviates Liver Injury in the Early Stages of Reperfusion

To elucidate the unique role of myeloid Trem2, which is recruited to respond to inflammation in hepatic ischemia-reperfusion injury, we respectively intervened with sterile liver IR surgery in Trem2*^FL/FL^* and Trem2*^mKO^* mice. We found that compared with Trem2*^FL/FL^* mice, Trem2*^mKO^* mice appeared to have reduced inflammatory damage at 6 h, manifested by lower serum ALT and AST levels (Figure 2A) and reduced histological damage (Figure 2B,C). At the same time, the transcription of the pro-inflammatory factors IL6, TNF, and iNOS activated by the IR program was also inhibited (Figure 2D). The dogmatic view is that macrophage Trem2 suppresses inflammation and protects the liver from injury [15], but this is contrary to our findings because blocking myeloid-derived Trem2 early in the induction of inflammation alleviates liver IR injury. The infiltration of LY6G+ neutrophils into the IR6h livers of myeKO mice was significantly reduced (Figure 2E). S100A9 is a calprotectin expressed in neutrophils and monocytes, and its expression is upregulated during acute or chronic liver inflammation [18,19]. Infiltration of CCL2+S100A9+ inflammatory monocytes and CCL2−S100A9+ neutrophils in the Trem2*^mKO^* IR liver was inhibited (Figure 2F).

### 2.3. Mitochondrial Damage and ROS Accumulation Are Significantly Reduced in Trem2-Depleted Macrophages

ROS levels in NPCs isolated from Trem2*^mKO^* livers at IR6h were significantly lower than those in Trem2*^FL/FL^* livers(Figure 3A), and the mitochondrial membrane potential and mitochondrial dysfunction also indicated that Trem2*^mKO^* indeed attenuated macrophage responses to inflammation and mitochondrial damage upon oxidative stress stimulation(Figure 3B,C). Importantly, analysis of the relative copy numbers of D-Loop/Tert and Non-Numt/β2M indicated that Trem2*^mKO^* BMDMs accumulated less mtDNA than WT BMDMs(Appendix A). In addition, the downregulation of mitochondrial genes in response to H/R oxidative stress injury was almost completely reversed in the Trem2-KO BMDMs (Appendix A). These results suggest that deletion of myeloid Trem2 decreases mtDNA accumulation in BMDMs. By detecting the mitochondrial membrane potential of BMDMs stimulated with LPS or LPS+IFN-γ in the presence or absence of Trem2 in vitro, we found that Trem2*^mKO^* macrophages also had less mitochondrial damage in response to IFN+LPS stimulation (Figure 3D,G). ROS production was also significantly reduced in Trem2*^mKO^* macrophages (Figure 3E,F).

### 2.4. Myeloid Trem2 is Responsible for Driving the Phenotypic Transformation of Monocyte-Derived Macrophages

In acute liver injury induced by APAP, Trem2 was shown to be responsible for the emergence of restorative macrophages during the resolution phase of inflammation, but global knockout obscured the characteristic role of myeloid Trem2 in this process [15] because Trem2 also plays an important role in Kupffer cells and dendritic cells during sterile inflammation [20,21]. Based on this, we aimed to elucidate the unique role of myeloid Trem2 in regulating the phenotypic reprogramming of macrophages in response to hepatic ischemia-reperfusion injury. Surprisingly, the absence of myeloid Trem2 had a differential direction of regulation of IL10 during the induction and resolution phases of inflammation (Appendix A). Mice overexpressing Trem2 were induced with tamoxifen to determine the induction efficiency by detection of the TdTomato signature by flow cytometry (Figure 4A). Mice overexpressing myeloid Trem2 had a significantly increased proportion of CD11b+CX3CR1hi macrophages relative to CD11b+CX3CR1low macrophages at 24 h post-IR (Figure 4B), indicating that overexpression of bone marrow Trem2 can promote CX3CR1 phenotype acquisition more quickly. In addition, CX3CR1hi macrophages of CCR2+/− or Ly6C+/− mice were significantly increased relative to those of CX3CR1low macrophages(Figure 4B). However, overexpression of Trem2 does not appear to accelerate IR-induced Ly6C and CCR2 phenotype resolution (Figure 4C). Analysis of the phenotype shows that myeloid Trem2 promotes the reprogramming of pro-resolution macrophages in the IR liver (Figure 4A–C). Trem2mKO IR24h-Livers infiltrated CX3CR1−Mφ rather than CX3CR1+Mφ(Figure 4D–F), which was accompanied by the same changes in the transcriptional level of CX3CL1 (Figure 4G). In addition, the deletion of Trem2 affected the successful expression of Arg1 (Figure 4H) by macrophages in the transition state after IR, thereby preventing M2-like macrophages from exerting an effective pro-resolution effect.

### 2.5. Myeloid Trem2 is Essential to Mφ Efferocytosis on the Stage of Liver IR Inflammation Resolution

A high-salt diet can weaken macrophage efferocytosis dependent on Trem2 after ischemic stroke, thereby aggravating neuroinflammation [22]. Therefore, we investigated whether peripheral Trem2 originating from myeloid cells was involved in promoting the resolution of inflammation in the liver by driving the efferocytosis of MoMφ. It is clear that more TUNEL-labeled apoptotic cells that were not successfully cleared accumulated in the livers of Trem2*^mKO^* mice during the 24 h resolution period after liver IR (Figure 5A,B). To clarify whether the accumulation of apoptotic cells was due to the impairment of efferocytosis by Trem2*^mKO^*, we incubated PHrodoSE-labeled mouse thymocytes previously induced by dexamethasone with BMDMs from Trem2*^FL^*^/*FL*^ or Trem2*^mKO^* mice to examine the effect of the lack of Trem2 on BMDMs efferocytosis. In the absence of IL4, the efficiency of BMDMs efferocytosis lacking Trem2 was significantly reduced (Figure 5C,D). Subsequently, we added IL4 in advance to drive BMDMs polarization to the anti-inflammatory state before co-incubating the two. 24 h pre-induction of IL4 significantly increased the phagocytic efficiency of Trem2*^FL/FL^* mice but also increased that of Trem2*^mKO^* mice. However, BMDMs from Trem2*^mKO^* mice did not reach the phagocytic efficiency of control BMDMs (Figure 5E,F). In addition to peripherally derived macrophages, peritoneal macrophages (pMφ) have also been identified as an important source of macrophages that regulate the resolution phase of aseptic liver injury [23]. We highlighted that the loss of myeloid Trem2 impairs the phenotypic conversion of monocyte macrophages, which makes it difficult to define the phagocytic efficiency of the two groups of macrophages at the same time point. To solve this difficult problem, we measured the efferocytosis ability of pMφ in peritoneal lavage fluid 11 days after zymosan injection into the abdominal cavity. The identity of peritoneal macrophages was clearly clustered by single-cell sequencing [24]. In fact, there was no significant difference in the phenotype of inflammation-associated macrophages between Trem2*^FL/FL^* and Trem2*^mKO^* mice at this time point. It is worth noting that the negative effect of Trem2*^mKO^* on efferocytosis is still significant (Figure 5G–I) at this time. Moreover, the clearance of neutrophils expressing Ly6G was significantly impaired compared to that in Trem2*^FL/FL^* mice (Figure 2E).

### 2.6. Trem2 is Involved in the Recognition and Internalization of Apoptotic Cells

How are macrophage Trem2s on different differentiation pathways derived from myeloid cells participate in efferocytosis programming? Macrophages need to recognize apoptotic cells through a keen sense of smell and then phagocytize and internalize the corpse. Usually, the corpse may be much larger than the phagocyte [25], such as when comparing apoptotic hepatocytes with Kupffer cells. Therefore, strong internalization is important. Cytochalasin D is an actin polymerization inhibitor that prevents efferocytosis [26]. We incubated BMDMs with or without Trem2 with cytochalasin D for 30 min before phagocytizing apoptotic thymocytes and then added apoptotic cells labeled with PHrodoSE. Only 1 μM of cytochalasin D can significantly inhibit the efferocytosis of BMDMs, whether stimulated with IL-4 in advance or not (Figure 6A). Unexpectedly, the efferocytosis efficiency of Trem2*^mKO^* BMDMs was further reduced compared with Trem2*^FL/FL^* BMDMs and was almost completely inhibited by 1 μM cytochalasin D (Figure 6A,B). The addition of 10 μM cytochalasin D completely blocked the operation of efferocytosis (Figure 6A). This demonstrates that the role of Trem2 may not only be limited to actin polymerization at the endocytic stage but may also be important for the recognition of apoptotic cells. In fact, we demonstrated in another study that monocyte-macrophage Trem2 is necessary for the recognition of apoptotic cell-oxidized lipid epitopes. Therefore, in order to further clarify that Trem2 is also indispensable for the recognition of apoptotic cells, we observed the efferocytosis activity of Raw264.7 cells transfected with the lentivirus shTrem2 and Trem2*^FL/FL^* or Trem2*^mKO^* BMDMs at 4 °C for 1.5 h and then observed by confocal microscopy. After incubation at 4 °C, the apoT labeled by CellTracker Blue aggregated around the macrophages but could not be internalized. BMDMs or Raw264.7 at this time still expressed Trem2 (Appendix A). BMDMs or Raw264.7 lacking expression of Trem2, attached less apoT labeled by cyan dye (Figure 6C–G). Overall, myeloid macrophage Trem2 is involved in both recognition and endocytosis of command efferocytosis.

### 2.7. Myeloid Trem2 Regulates Cox2/PGE2-Mediated Rac1 Activation in Efferocytic Mφs

Phalloidin-labeled actin was significantly reduced, and phagocytic cup formation was also inhibited by immunofluorescence microscopy in Trem2*^mKO^* BMDMs when the efferocytosis procedure was run (Appendix A), F-actin activation of Lenti-shTrem2 transfected Raw264.7 efferocytosis also reduced (Appendix A). The upregulation of Cox2 in the liver during IR resolution was reversed by the absence of myeloid Trem2 (Appendix A). PGE2 is an important pro-resolving mediator for the completion of M2 polarization and efferocytosis in macrophages [27]; its expression was detected in the liver and efferocytosis supernatant during IR resolution (Figure 7A,B) but was inhibited by Trem2*^mKO^* (Figure 7A,C). Its activation during efferocytosis is regulated by Cox2 [27,28]. Treatment with the selective inhibitor, celecoxib, impaired efferocytosis in mouse primary macrophages (Appendix A). In addition, the expression of Rac1 is also essential for the phagocytosis of apoptotic cells [29,30]. It has been shown that PGE2 is responsible for driving Rac1-related efferocytosis [31]. The application of the Cox2 inhibitor celecoxib to IR mice alone inhibited PGE2 activation, similar to Trem2*^mKO^*. However, further use of celecoxib based on Trem2*^mKO^* did not further reduce PGE2 production or Rac1 expression (Figure 7C,H). The transcriptional level and protein activity of upregulated Rac1 in the liver 24 h after IR were reversed by Trem2*^mKO^* (Figure 7D–F), consistent with the decreased Rac1 protein activity in shTrem2-Raw264.7 subjected to efferocytosis program in vitro; Rac1 mainly drives actin Remodeling to help macrophages complete the phagocytic program. Trem2*^mKO^* resulted in the inhibition of efferocytosis-related Rac1 activation (Figure 7G); we further treated efferocytosis-treated BMDMs with phalloidin after lentiviral treatment with shRac1 on the basis of Trem2*^mKO^* and F-actin aggregation was not observed in Trem2*^mKO^* was further inhibited (Figure 7I,J); the efficiency of efferocytosis was also not further impaired on the basis of shRac1 (Figure 7K). Collectively, myeloid Trem2-dependent Cox2/PGE2-mediated Rac1-regulated actin polymerization promoted the internalization of apoptotic cargo.

## 3. Discussion

Our study revealed that myeloid Trem2 might play a bidirectional role at different stages of inflammation in response to different ligands and mediators. In the induction phase of inflammation, deletion of myeloid Trem2 inhibits the activation of iNOS, MCP-1, and CXCL1/2 (Figure 2), which in turn leads to a reduction in the recruitment of inflammatory monocytes and S100A9+ neutrophils and production of ROS. Notably, the deletion of myeloid Trem2 also affected the reprogramming of inflammatory monocytes from Ly6C^hi^CX3CR1_low_ to reparative macrophages from Ly6C_low_CX3CR1^hi^ and impaired the efficiency of efferocytosis by inhibiting the recognition of apoptotic cells and preventing macrophage actin polymerization driven by Rac1, thus delaying the inflammatory repair of liver IR. In summary, the absence of myeloid Trem2 attenuates the degree of induction of liver injury but simultaneously slightly impairs the timely resolution of the IR liver.

In recent years, the core role of myeloid cells in various pathological stress processes has gradually been recognized. Trem2, a transmembrane receptor of the immunoglobulin superfamily, is considered a major pathologically induced immune signal hub that can sense tissue damage and induce rapid activation of immune remodeling [32]. In vitro studies based on mouse macrophages showed that after Trem2 ligand interaction, DAP10/12 was phosphorylated as a co-receptor, and intracellular signal transduction was recruited. DAP12 (TYROBP) mediates the phosphorylation of the spleen tyrosine kinase Syk, whereas DAP10 propagates through the recruitment of PI3K cascade signals [32]. Trem2 is derived from circulating monocytes, expressing newly differentiated and alternatively activated macrophages, and inhibits macrophage activation. The presence of Trem2 inhibits the response of macrophages to LPS, yeast glycans, and CpG. In fact, we showed that IL1β from BMDMs lacking Trem2 responded more strongly to LPS stimulation, but mitochondria and tissue damage were indeed alleviated in the whole liver in response to IR, suggesting that the effect of Trem2 on oxidative stress may be greater. Although the lack of Trem2 upregulates the transcription of IL1β stimulated by LPS, there is additional regulation of the cleavage of IL1β. Theoretically, the discrepancy between our findings and the dogma of Trem2 regulation of inflammation and tissue injury may arise from a specific myeloid knockout mouse construct, ignoring the identity of Trem2 in liver-resident macrophages and dendritic cells.

The induction and resolution of hepatic aseptic inflammation are well-programmed cascade processes. In this process, peripherally derived mononuclear macrophages (MoMFs) play a regulatory role in cooperation with a host of immune cells, including Kupffer cells and resident DC, to ensure the steady-state maintenance of the liver in the event of severe injury [33,34]. CCR2hiCX3CR1low monocytes are the only source of peripheral infiltrating pro-inflammatory macrophages in the early stages of aseptic liver injury. Ly6C+ monocytes are gradually programmed into CX3CR1+ repair macrophages in response to stimulation by cytokines such as IL10 [35]. Monocyte macrophages are also involved in promoting the repair of liver inflammation because pre-depletion of CD11b+ MoMF alleviates liver cell damage induced by CCL4, while depletion of CD11b+ MoMF in the recovery phase delays tissue repair [4]. The myeloid deletion of Trem2 resulted in a delay in macrophage reprogramming during IR, which prolonged the “window period” for Ly6C+CX3CR1+ macrophages (Figure 4).

Therefore, how do monocyte macrophages expressing Trem2 regulate the resolution of liver IR inflammation? The resolution of aseptic inflammation is synergistically promoted by the efferocytosis of macrophages from different sources [36,37]. It can be speculated that efferocytosis of monocyte macrophages in the resolution phase is very important because IR leads to the depletion of Kupffer cells [38]. In the liver, Kupffer cell TIM4 promotes efferocytosis by upregulating IL10, thereby helping resolve the inflammation caused by IR injury [7]. Exogenous netrin-1 treatment enhances the efferocytosis of macrophages into apoptotic PMN, thus promoting the protection and repair of the liver [39]. Similarly, macrophage MerTK is equally important for tissue repair after efferocytosis-dependent myocardial injury and aseptic peritonitis [40,41]. Macrophage efferocytosis also promotes the resolution of myocarditis by inducing VEGFC production [42]. Regulation of inflammation in JAML after renal ischemia-reperfusion injury also depends on efferocytosis [43]. MerTK and Arg1 are essential for the polarization and efferocytosis of M2 [26,44]. Thus, Trem2 deficiency impairs the efferocytosis program whether stimulated with IL4 or not (Figure 5). Mechanistically, efferocytosis relies on actin polymerization and phagocytic cup formation to complete phagocytosis and internalization. The expression of Rac1 is essential for the completion of phagocytosis by apoptotic cells [29,30]. The lack of Trem2 leads to the inhibition of Rac1-related actin polymerization, thus preventing the successful internalization of efferocytosis (Figure 6 and Figure 7). COX2/PGE2 activity during the resolution of hepatic IR is directly regulated by myeloid Trem2 and contributes to Rac1-related endocytic signaling [27,28] (Figure 7). Overall, myeloid Trem2 is important for mononuclear macrophages to complete efferocytosis, which promotes regression of hepatic IR inflammation.

## 4. Materials and Methods

### 4.1. Animals

We housed Trem2*^FL/FL^*, Trem2*^mKO^*, and Trem2*^creERT^*Rosa26*^TdTomato^* mice in ventilated rack caging in a pathogen-free, constant-humidity SPF-grade animal facility at room temperature with a 12-h light/12-h dark cycle. Trem2*^Flox^* (Trem2*^FL/FL^*) mice (Shanghai Model Organisms, Shanghai, China) and mice expressing Cre recombinase under the control of the Lysozyme 2 (Lyz2) promoter (LysM-Cre; Shanghai Model Organisms, Shanghai, China) were used to generate myeloid-specific Trem2 knockout (Trem2*^mKO^*) mice. Male wild-type C57BL/6 mice (6–8 weeks old) were purchased from Vital River Laboratory Animal Technology Co., Ltd. (Beijing, China). All murine experiments were performed under the supervision of the Animal Ethics Committee of Nanjing Medical University (IACUC-2102039).

### 4.2. Mouse Liver Partial Warm Ischemia Model

Male mice aged 6–8 weeks were subjected to 90-min of ischemia, then 6 h to 24 h reperfusion. Briefly, under isoflurane anesthesia, a midline laparotomy was performed under isoflurane anesthesia. The mice were then injected with heparin (100 U/kg), and an atraumatic clip was used to interrupt both the arterial and portal venous blood supply to the cephalad-liver lobes. After 90-min of partial hepatic ischemia, the clip was removed to initiate hepatic reperfusion. The sham-operated controls underwent the same procedure but without vascular occlusion. Mice were euthanized for 3 h–7 days reperfusion to obtain liver and serum samples. Serum alanine aminotransferase (sALT) and aspartate aminotransferase (sAST) were measured. Liver specimens were fixed in 4% neutral buffered formalin and then embedded in paraffin. Liver sections (4 μm thick) were stained with hematoxylin and eosin (H&E) or antibodies using standard immunohistochemistry protocols.

### 4.3. Liver NPC Isolation

Liver NPCs were isolated from mice by in situ liver perfusion. Briefly, the liver was perfused in situ via the portal vein with calcium- and magnesium-free HBSS supplemented with 2% heat-inactivated FBS. This was followed by 0.27% collagenase IV (Sigma, St. Louis, MO, USA). Perfused livers were dissected and teased through 70 μm nylon mesh cell strainers (BD Biosciences, San Diego, CA, USA). Non-parenchymal cells (NPCs) were separated from hepatocytes by centrifuging at 50× *g* for 3 min three times. NPCs were stained with fluorescence-labeled antibodies and analyzed using FACS.

### 4.4. Measurement of mtDNA

Using qPCR, mtDNA content was determined as the ratio of D-loop mtDNA to Tert nuclear (n) DNA or (B) mtDNA that was not inserted into nuclear DNA (non-NUMT) to B2m nDNA.

### 4.5. Measurement of ROS and Mitochondrial Membrane Potential

Macrophage ROS levels were detected using a reactive oxygen species assay kit (Byotime, S0033M, Shanghai, China). Briefly, a serum-free working solution of ROS at a concentration of 1 mM was prepared and then added to the dish to incubate BMDMs at 37 °C for 30 min, then detected by fluorescence microscopy or flow cytometry.

The mitochondrial membrane potential was detected using an enhanced mitochondrial membrane potential assay kit with JC1(Beyotime, C2003S, Shanghai, China). A working solution of JC1 was prepared at a concentration of 3 mM, added to BMDMs at 106/mL medium, and incubated at 37 °C for 30 min, and the mitochondrial membrane potential was detected by flow cytometry.

### 4.6. Lentiviral Preparation of shTrem2 and shRac1

shTrem2 and shRac1 lentiviruses were constructed using PGMLV-hU6-MCS-CMV-ZsGreen1-PGK-Puro-WPRE as a vector, and the infection virus titer was 1E8TU. The infection efficiency against Raw264.7 was determined by observing the expression of green fluorescent cells by fluorescence microscopy.

### 4.7. Mass Spectrometry Analysis

Mass spectrometry analysis of the abundance of COX2 and PGE2 in mouse livers 24 h after IR was performed using the AB Sciex QTRAP 6500 LC-MS/MS platform.

### 4.8. Extraction and Culture of Bone Marrow-Derived Macrophages (BMDMs)

Bone marrow cells were isolated from the femurs and tibias of mice. Cells were filtered through a 70 μm nylon mesh cell strainer (BD Biosciences, San Diego, CA, USA) and lysed with red blood cell lysis to remove red blood cells. Cells were cultured in DMEM supplemented with 10% FBS, 20% L929-conditioned medium, and 100 U/mL penicillin-streptomycin for 7 days.

### 4.9. Induction and Isolation of Peritoneal Macrophages

For the isolation of peritoneal macrophages, mice were injected intraperitoneally with 1mL 4% zymosan. After 3 or 11 days, peritoneal cells were isolated by washing the peritoneal cavity with 5 mL PBS.

### 4.10. Isolation of Thymocytes

To isolate thymocytes, the thymus was harvested from euthanized mice using institution-approved protocols. Then thymocytes were prepared by gently grinding the organ between frosted glass slides. After passing through a 70 μm nylon mesh cell strainer (BD Biosciences, San Diego, CA, USA) and centrifugation, the isolated cells were resuspended by RBC lysis solution to remove red blood cells. Isolated thymocytes were cultured in complete DMEM (10% FBS, 100 U/mL penicillin-streptomycin, and 4 mM L-glutamine).

### 4.11. Preparation of Apoptotic Thymocytes

Apoptotic thymocytes (10^7^/mL) were obtained by incubation with dexamethasone (0.1 μM) in complete DMEM for 16 h.

### 4.12. Efferocytosis Assay

Apoptotic thymocytes were labeled with 20 ng/mL pHrodo red (Invitrogen, CA, USA) for 30 min. 10^6^ BMDMs or peritoneal macrophages were co-cultured with 4 × 10^6^ pHrodo-SE-labeled apoptotic thymocytes. After incubating for 1.5 h or 2.5 h, the cells were detached from the plate, transferred to FACS tubes, and analyzed by flow cytometry. Alternatively, adherent cells were visualized using confocal fluorescence microscopy.

### 4.13. Immunohistochemistry and Immunofluorescence

For Immunohistochemistry, liver tissue samples were fixed in 4% paraformaldehyde, embedded in paraffin, and sliced into 4 μm thick sections for hematoxylin-eosin (H&E) staining. For immunofluorescence, the fixed tissue sections were placed in EDTA antigen retrieval buffer (PH 9.0) and then boiled in a microwave for repair. BSA (3%) in PBS solution was added for 30 min to block nonspecific binding, and then the corresponding primary antibodies anti-Trem2 (ab245227, Abcam, MA, USA) and anti-F4/80 (ab6640, Abcam, MA, USA) were added. After overnight incubation, the secondary antibody of the corresponding species was added for 50 min in the dark, and the nuclei were stained. Slides were observed under a confocal fluorescence microscope (NIKON ECLIPSE C1, Tokyo, Japan).

### 4.14. Flow Cytometry

BMDMs or peritoneal macrophages were isolated from mice as described above. A total of 1 × 10^6^ cells were incubated with Fc receptor CD16/32 blockers for 10 min and then suspended in a cell staining buffer (420201, BioLegend, San Diego, CA, USA). To recognize pHrodo-SE and CellTracker Blue carried by apoptotic cells to determine efferocytosis efficiency, no additional antibodies were required. For surface antibodies, the cells were stained with anti-F4/80 (123110, BioLegend, San Diego, CA, USA), anti-CD11b-FITC (101206, BioLegend, San Diego, CA, USA) for 20 min and then subjected to flow cytometry analysis (CytoFLEX, Beckman, California, CA, USA). Raw data were analyzed using FlowJo (10.8.1).

### 4.15. Quantitative RT-PCR

After extraction from tissues or cells, total RNA (2.0 μg) was reverse-transcribed into cDNA using SuperScriptTM III First-Strand Synthesis System (Invitrogen, Carlsbad, CA, USA). Quantitative PCR was performed using a DNA Engine with a Chromo 4 Detector (MJ Research, Waltham, MA, USA). In a final reaction volume of 20 μL, the following were added: 1xSuperMix (Platinum SYBR Green qPCR Kit, Invitrogen, Carlsbad, CA, USA), cDNA, and 0.5 mM of each primer. Primer sequences used to amplify Trem2, TNF-a, iNOS, IL-6, IL-10, CX3CL1, Arg1, and Cox2 are listed in Appendix A. All expression levels and results of the target genes were standardized for β-actin expression.

### 4.16. ELISA Assay

Murine serum and cell-free culture supernatants were collected. Then an ELISA kit (R&D Systems, MND, USA) was used to test levels of IL-10 and PGE2.

### 4.17. Western Blot

Tissues or cells were lysed with RIPA, separated by SDS-PAGE gel electrophoresis, and transferred to a PVDF membrane. The proteins were immunoblotted with anti-Trem2 (61788S, Cell Signaling Technology, BSN, USA) and anti-β-actin (4970, Cell Signaling Technology, BSN, USA) as primary antibodies. This was followed by color development using horseradish peroxidase (HRP)-enhanced chemiluminescence (ECL) assays.

### 4.18. Statistical Analysis

Representative results are shown, and data are presented as mean ± SD. Statistical significance was analyzed using Student’s unpaired *t*-test or one-way ANOVA followed by Bonferroni’s method for post hoc pairwise multiple comparisons. Two-tailed *p*-values less than 0.05 were considered statistically significant.

## 5. Conclusion

In conclusion, our study demonstrates a complex dynamic role for myeloid-derived Trem2 in hepatic ischemia-reperfusion injury. While expounding its specific role and relevant regulatory mechanism in the induction and resolution period of hepatic aseptic inflammation, we explored the limitations of targeting myeloid Trem2 to treat an aseptic liver injury and provided ideas for clinical improvement of hepatic ischemia-reperfusion injury.

## Figures and Tables

**Figure 1 ijms-24-06348-f001:**
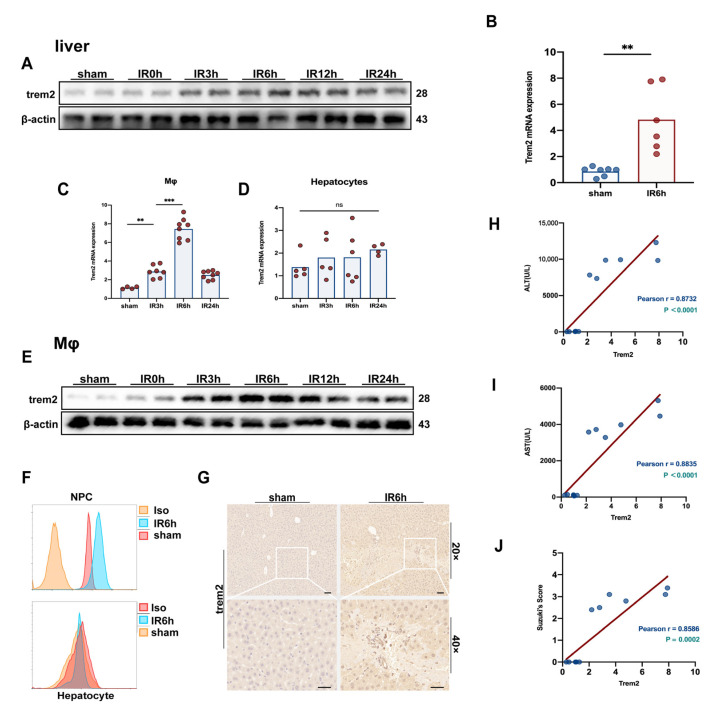
Macrophages Trem2 was activated in the IR liver. The protein and mRNA expression of Trem2 in the liver at different time points after IR was detected by (**A**) Western blot and (**B**) qPCR (*n* = 6–7/group, ** *p* < 0.01); (**C**,**D**) Trem2 mRNA levels in NPCs or hepatocytes isolated from IR liver were detected by qPCR (*n* = 4–8/group, ns = Not Significant, ** *p* < 0.01, *** *p* < 0.001); The expression of Trem2 protein in NPC and hepatocyte isolated from IR liver was detected by (**E**) Western blot or (**F**) Flow cytometry. (**G**) The expression of Trem2 was detected by IHC, scale bars = 50 μm; (**H**,**I**) Correlation between serum ALT/AST and Trem2 mRNA expression; (**J**) Correlation between liver Suzuki’s score and Trem2 mRNA expression.

**Figure 2 ijms-24-06348-f002:**
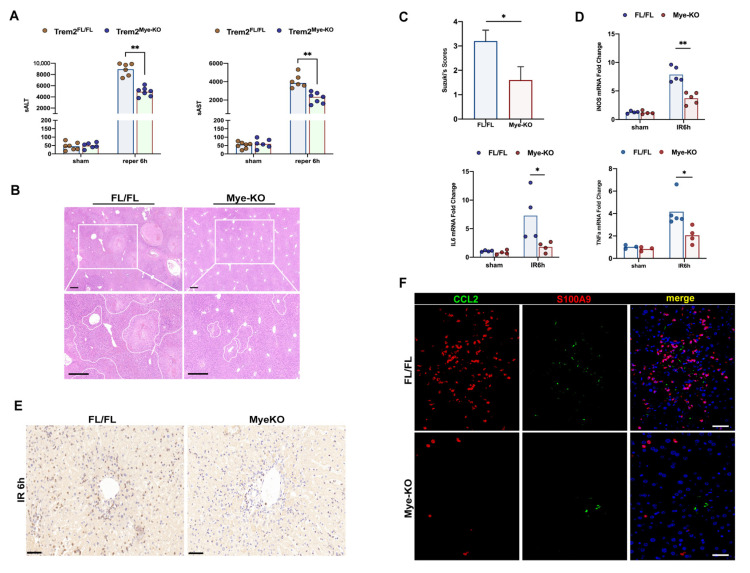
Blockade of myeloid Trem2 attenuated the induction of liver IR injury. (**A**) The levels of serum ALT and AST at sham and IR6h in mice with Trem2*^FL/FL^* or Trem2*^mKO^* were detected (*n* = 6–7/group, ** *p* < 0.01); (**B**) HE staining was used in determining the degree of liver tissue damage caused by IR in Trem2*^FL/FL^* or Trem2*^mKO^*, scale bars = 500 μm; (**C**) Liver Suzuki’s score was used to analyze the liver tissue injury of Trem2*^FL/FL^* or Trem2*^mKO^*(* *p* < 0.05, ** *p* < 0.01); (**D**) Expression of IL6, iNOS, and TNFα in IR6h liver of Trem2*^FL/FL^* or Trem2*^mKO^* by qPCR (*n* = 4–5/group, * *p* < 0.05, ** *p* < 0.01); (**E**) IHC of Ly6G analyzed neutrophil infiltration in IR livers of mice with Trem2*^FL/FL^* or Trem2*^mKO^*, scale bars = 50 μm; (**F**) CCL2 and S100A9 were used to analyze the infiltration of immunosuppressive neutrophils in the IR liver of mice with Trem2*^FL/FL^* or Trem2*^mKO^*, scale bars = 100 μm.

**Figure 3 ijms-24-06348-f003:**
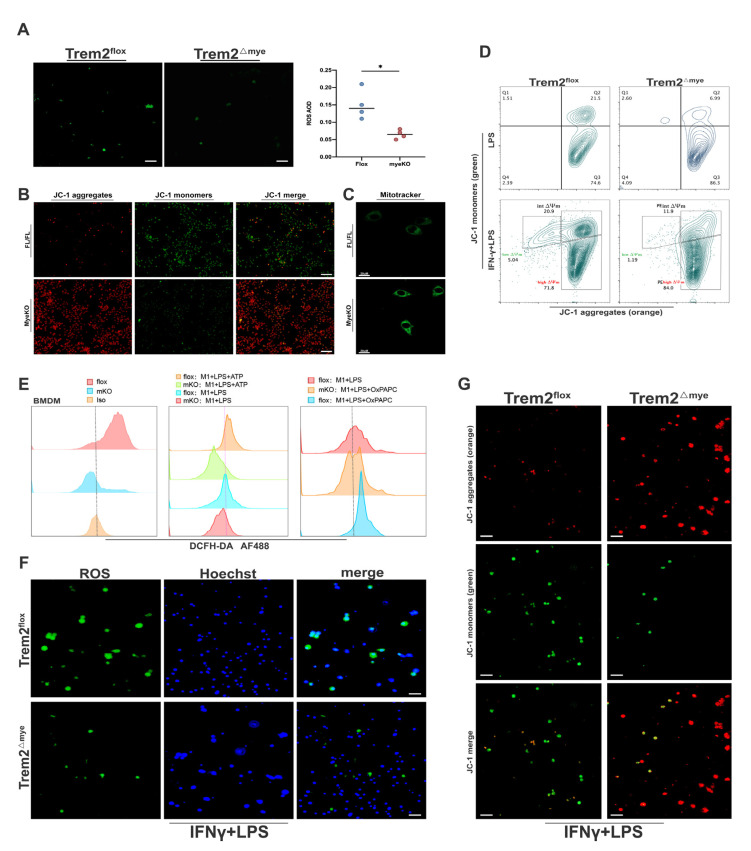
Trem2 deficiency suppresses mitochondrial damage, and ROS release in macrophages in response to oxidative stress (**A**) ROS production from liver NPC of Trem2*^FL/FL^* or Trem2*^mKO^* at IR6h was detected by immunofluorescence (* *p* < 0.05, scale bars = 100 μm); (**B**) JC1 detects mitochondrial membrane potential (scale bars = 200 μm) and (**C**) Mitotracker detects mitochondrial function (scale bars = 20 μm). (**D**) Mitochondrial membrane potential of BMDMs by 1 μg/mL LPS stimulated for 6 h or 200 ng/mL LPS + 10 ng/mL IFNγ for 24 h and then measured by JC1 enhanced detection kit; (**E**) ROS release was measured by flow cytometry DCFH-DA; (**F**) Mitochondrial membrane potential, scale bars = 200 μm and (**G**) ROS changes of BMDMs with or without Trem2 in response to LPS or IFNγ + LPS stimulation was detected by immunofluorescence, scale bars = 200 μm.

**Figure 4 ijms-24-06348-f004:**
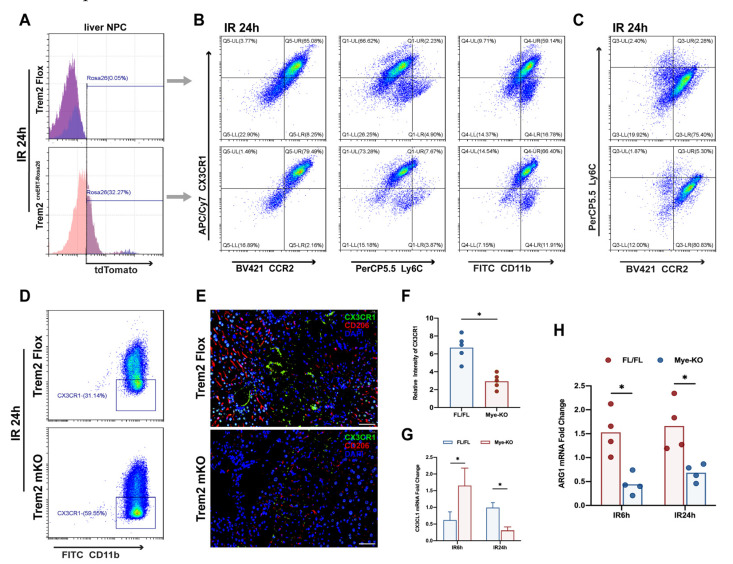
Reprogramming of mono-derived macrophages depends on myeloid Trem2. After 5 days of intraperitoneal injection of tamoxifen into Trem2*^creERT^*Rosa26*^Tdtomato^* mice and Trem2*^Fl/FL^* mice, a liver IR operation was performed. After 24 h, liver NPCs were isolated, and the (**A**) overexpression efficiency of Tdtomato labeling was verified by flow cytometry. (**B**,**C**) the expression of CX3CR1, Ly6C, CCR2, and CD11b in the liver NPCs of Trem2*^creERT^* Rosa26*^Tdtomato^* and Trem2*^Fl/FL^* in IR24h was detected by flow cytometry; (**D**) The expression of CX3CR1 in the liver of Trem2*^FL/FL^* or Trem2*^mKO^* at IR24h was detected by flow cytometry; (**E**) The expression of CX3CR1 and CD206 in Trem2*^FL/FL^* or Trem2*^mKO^* livers was detected by immunofluorescence (scale bars = 100 μm) and (**F**) the fluorescence intensity was quantified by ImageJ (*n* = 5/group, * *p* < 0.05); The mRNA levels of (**G**) CX3CL1 and (**H**) Arg1 in the livers of Trem2*^FL/FL^* or Trem2*^mKO^* at IR24h were detected by qPCR (*n* = 4–5/group, * *p* < 0.05).

**Figure 5 ijms-24-06348-f005:**
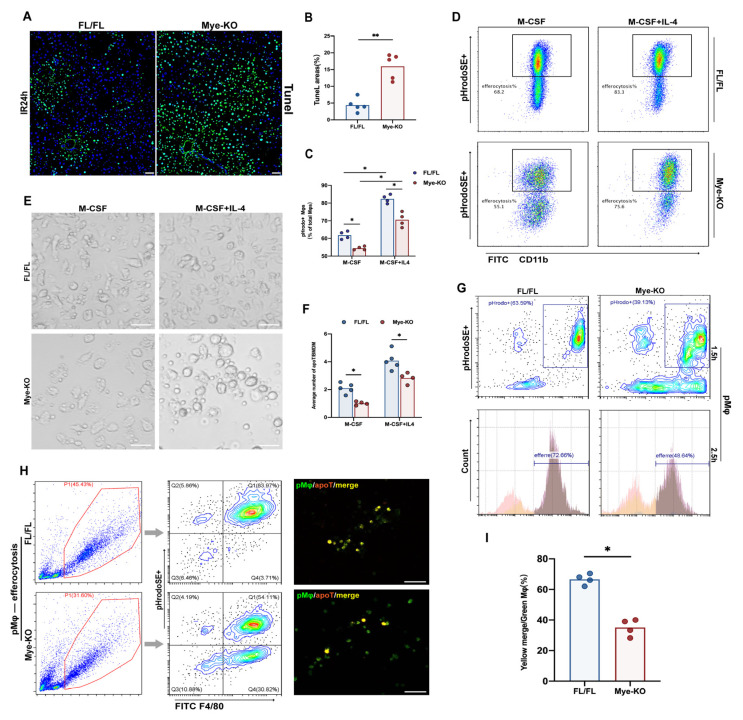
Myeloid Trem2-controlled efferocytosis promotes the resolution of IR inflammation. (**A**) Tunel staining was used to determine the number of apoptotic cells in the IR24h liver of Trem2*^FL/FL^* or Trem2*^mKO^*, scale bars=50 μm; (**B**) Qualification of Tunel staining (** *p* < 0.01); (**C**,**D**) BMDMs of Trem2*^FL^*^/*FL*^ or Trem2*^mKO^* were co-cultured with dexamethasone-induced thymocytes incubated with pHrodo in the presence or absence of IL4 for 2 h, and then the ratio of pHrodo+SE was detected by flow cytometry (* *p* < 0.05); (**E**,**F**) calculate the average number of apoptotic cells phagocytized by each macrophage (* *p* < 0.05, scale bars = 20 μm); (**G**) Peritoneal macrophages were isolated 3 days after intraperitoneal injection of zymosan into Trem2*^FL/FL^* or Trem2*^mKO^* mice and then co-cultured with dexamethasone-induced thymocytes incubated with pHrodo for 1.5 h or 2.5 h, and then the efferocytosis efficiency was detected; (**H**) Peritoneal macrophages were isolated 11 days after intraperitoneal injection of zymosan into Trem2*^FL/FL^* or Trem2*^mKO^* mice, and then co-cultured with dexamethasone-induced thymocytes incubated with pHrodo for 1.5 h,scale bars = 50 μm. (**I**) Then the efferocytosis efficiency was detected (* *p* < 0.05).

**Figure 6 ijms-24-06348-f006:**
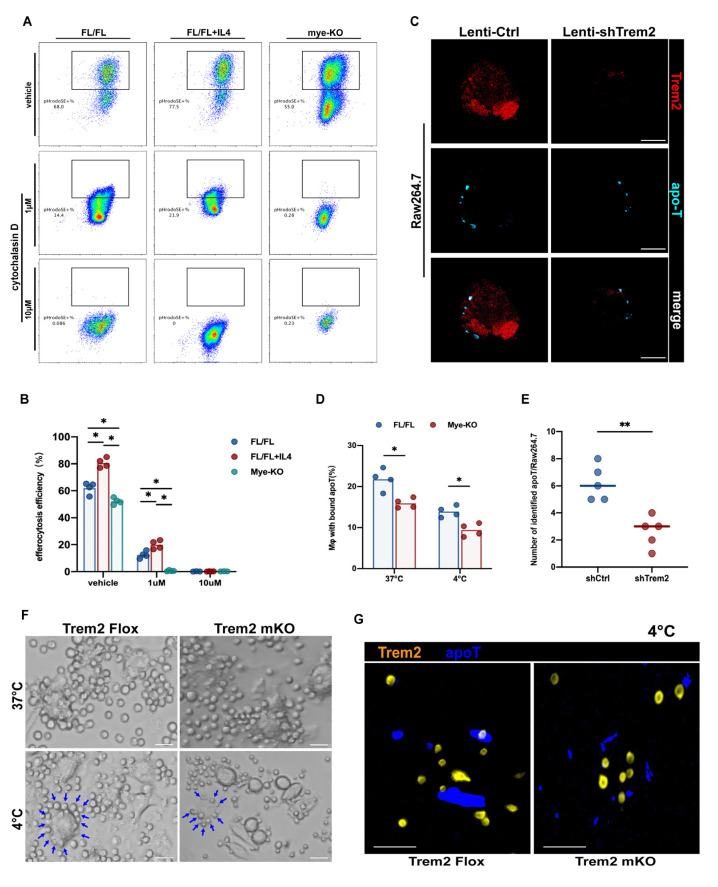
Trem2 is involved in regulating the recognition and internalization of efferocytosis by monocyte macrophages. (**A**) BMDMs under different stimulation conditions, such as Trem2*^FL^*^/*FL*^, Trem2*^FL^*^/*FL*^+IL4, and Trem2*^mKO^*, were incubated with 1 μm or 10 μm cytochalasin D when running efferocytosis, and then (**B**) the efficiency of efferocytosis was detected (*n* = 4, * *p* < 0.05); (**C**,**E**) the number of apoptotic thymocytes combining with Raw264.7 per Lenti-shCtr1 or Lenti-shTrem2 at 37 °C or 4 °C (*n* = 4, ** *p* < 0.01, scale bars = 100 μm); (**D**,**F**,**G**) Number of BMDMs bound apoptotic thymocytes per Trem2*^FL/FL^* or Trem2*^mKO^* at 37 °C or 4 °C. The blue arrows represent the apoptotic thymocytes attached to BMDMs (*n* = 5, * *p* < 0.05, scale bars = 10 μm).

**Figure 7 ijms-24-06348-f007:**
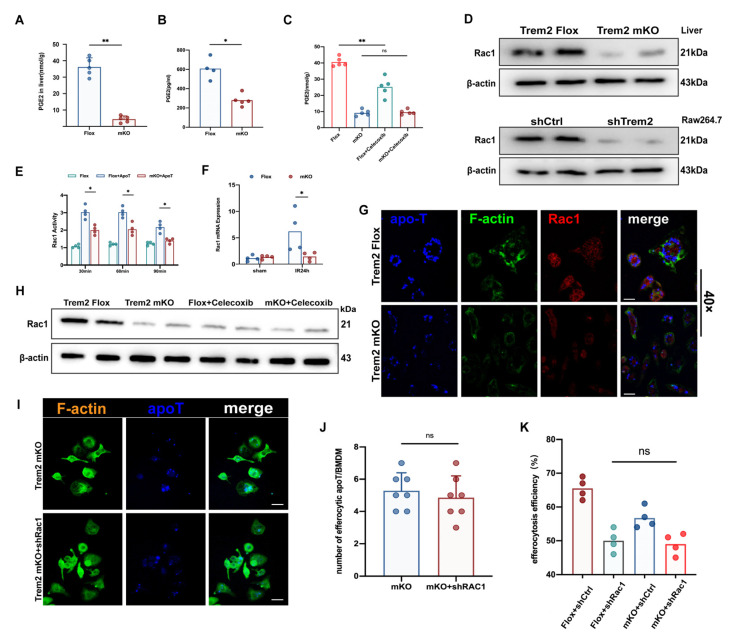
Trem2 conducts Rac1 signal in phagocytes through the COX2/PGE2 axis. (**A**) the relative abundance of PGE2 in the liver of Trem2*^FL/FL^* or Trem2*^mKO^* at IR24h was detected by mass spectrometry (nmol/g, *n* = 5, ** *p* < 0.01); (**B**) The PGE2 content in the supernatant of efferocytosis or (**C**) the liver of mice injected with Trem2*^FL/FL^* or Trem2*^mKO^* intraperitoneally with celecoxib for 24 h was detected (*n* = 4–5, ns = Not Significant,* *p* < 0.05, ** *p* < 0.01); (**D**) The protein expression of Rac1 in Trem2*^FL/FL^* or Trem2*^mKO^* livers of IR24h and Rac1 in RAW264.7 of Lenti-shCtr1 or Lenti-shTrem2 subjected to efferocytosis were detected by Western blot; (**E**,**G**) the activity of Rac1 in Trem2*^FL/FL^* or Trem2*^mKO^* BMDMs at different efferocytosis times (scale bars = 100 μm) and (**F**) the expression of Rac1 in the liver at IR24h (*n* = 4, * *p* < 0.05); (**H**) Western blot was used to detect the expression of Rac1 in the livers of mice treated with Trem2*^FL/FL^* or Trem2*^mKO^* injected intraperitoneally with celecoxib for 24 h; (**I**–**K**) Identification of phagocytic efficiency and F-actin remodeling when BMDMs with Trem2*^mKO^* or Trem2*^mKO^* + shRac1 exercised the efferocytosis program by immunofluorescence (scale bars = 50 μm, ns = Not Significant).

## Data Availability

Not applicable.

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
