# Peer review of "Myeloid Trem2 Dynamically Regulates the Induction and Resolution of Hepatic Ischemia-Reperfusion Injury Inflammation"

_ijms, 2023, doi:10.3390/ijms24076348_

Round 1

Reviewer 1 Report

Han S. et al characterized, expression of Trem2 in both bone marrow and embryonic-derived macrophages in the liver and has a complex role in liver inflammation. In the early stages of inflammation induced by ischemia-reperfusion injury, myeloid Trem2 deletion inhibited the induction of iNOS, MCP-1, and CXCL1/2, reduced neutrophil accumulation, mitochondrial damage, and decreased ROS formation. However, in later stages, Trem2 promotes phagocytosis of apoptotic cells and the resolution of inflammation by controlling Rac1-related actin polymerization and regulating the Cox2/PGE2 axis. The role of Trem2 is complex and heterogeneous at different stages, contributing to understanding sterile inflammatory immunity and exploring regulatory strategies for targeting Trem2 in sterile liver injury. The manuscript is interesting and well written. Although authors need to address some major points to improve its quality:

1.    In fig. 1 did author check protein level of Trem2 at 12 h of IR in both hepatocytes and macrophages? I wonder the protein level of Trem2 looks peak at 6h, (it should be 12 h also) in both hepatocytes and macrophages, however it goes down at 24 h of IR

2.    Again, I wonder how author thinks that Trem2 level is peak at just 6 h of IR without checking 8 or 12h of IR? I urge author to check the serum ALT and AST levels in TREM2FL/FL and Trem2mye mice also at different time points (i.e 0,6,12,24h of IR)

3.     Would you please describe how you measure ROS level in materials and method section? Also what is the source of ROS induced in TREM2FL/FL mice? Is it from mitochondria?

Author Response

Response to Reviewer 1 Comments

Point 1:In fig. 1 did author check protein level of Trem2 at 12 h of IR in both hepatocytes and macrophages? I wonder the protein level of Trem2 looks peak at 6h, (it should be 12 h also) in both hepatocytes and macrophages, however it goes down at 24 h of IR

Response 1: Combined with other people's research and our experiments, Trem2 protein is mainly expressed on macrophages, but not on hepatocytes(Fig.1F). Through Western Blot detection, we found that Trem2 in the whole liver and macrophages still had high protein expression at 12h after IR, but slightly decreased compared with 6h, and continued to decrease at 24h after IR (Fig.1A, 1E), but still activated. Notably, the decrease in Trem2 expression was more pronounced in the whole liver than in macrophages, which may be due to the depletion of Kupffer cells (Fig. 1E).

Point 2: Again, I wonder how author thinks that Trem2 level is peak at just 6 h of IR without checking 8 or 12h of IR? I urge author to check the serum ALT and AST levels in TREM2FL/FL and Trem2△mye mice also at different time points (i.e 0,6,12,24h of IR)

Response 2: We detected Trem2 mRNA expression in sham, 0h, 3h, 6h, 12h, 24h (Fig.1A,1E). According to the reviewer's suggestion, we additionally detected ALT and AST levels at 3h and 24h (Fig. 2A).

Point 3: Would you please describe how you measure ROS level in materials and method section? Also what is the source of ROS induced in TREM2FL/FL mice? Is it from mitochondria?

Response 3: Based on the reviewer's comments, we have added detailed methods on ROS measurement in the Methods section. To the best of our knowledge, ROS are mainly produced by damaged mitochondria of macrophages/hepatocytes and recruited neutrophils in the IR liver.

We appreciate for Editors and Reviewers’ warm work earnestly, and hope that the correction will meet with approval.

Once again, thank you very much for your comments and suggestions.

Thank you and best regards.

Sincerely,

Liyong Pu

E-mail:

[email protected]

Reviewer 2 Report

This research article offers interesting hints as to the role of TREM2 in the setting of ischemia-reperfusion (IR) of the liver. The Authors have provided several lines of evidence which support their statement that myeloid-derived TREM2 cells play a complex dynamic role in liver IR. In fact, this work shows that TREM2 may have contrasting functions, according to the timing of liver IR and other factors. The Authors try to present a comprehensive explanation for these seemingly contradictory findings, and support their thesis with data derived from several different experimental settings.

In general, I think that this manuscript should be published, as it adds novel information to the complex biology of TREM2.

However, I suggest a thorough revision of the manuscript for the English which was used. Several sentences are hard to read and should be written in a more understandable language. 

A few errors in capitalization or other format issues are present too (also in the supplementary figures).

Please find below a few points which may be improved: 

-l.34 -> “damp” should be capitalized

-l.35 -> the adjective “comprehensive” seems out of place 

-l.38 -> unclear sentence

-l. 39-> IRI is not defined

(in general, I suggest that the first paragraph in the Introduction should be wholly rewritten/revised)

-l.53 -> what is “APAP”?

-l.57 -> is “relevant” used as a synonym for “important”? If that is the case, I think that is an error

-l.64 -> “pro-inflammatory and anti-inflammatory”... triggers? Stimuli?

-l. 95-> NPC should be mentioned as an abbreviation of “non-parenchymal cells” the first time it appears in the text

-l.216-219 -> this sentence seems a little misplaced in the Result section, I suggest that the Authors move it in the Discussion

-l.243 -> does “H/R” stand for ischemia/reperfusion? If that is the case, acronyms should be used consistently throughout the manuscript

-Figure 3E is too small, even zooming in I could not read the legend

-l.279-281 -> unclear sentence

-l.340-343 ->  this sentence seems a little misplaced in the Result section, I suggest that the Authors move it in the Discussion, maybe rephrasing it with more technical language.

-Figure 6B should have significance bars also comparing FL/FL and MyeKO

Author Response

-l.34 -> “damp” should be capitalized

Response: Modified to DAMP

-l.35 -> the adjective “comprehensive” seems out of place 

Response: Removed “Comprehensive”

。。。

-l.38 -> unclear sentence

Response: Modified

-l. 39-> IRI is not defined

Response: Modified to ‘IR injury’

In general, we completely revised the first paragraph of the Introduction to make it more readable

-l.53 -> what is “APAP”?

Response: Added full name: n-acetyl-p-aminophenoln-

-l.57 -> is “relevant” used as a synonym for “important”? If that is the case, I think that is an error

Response: Modified

-l.64 -> “pro-inflammatory and anti-inflammatory”... triggers? Stimuli?

Response: We prefer to maintain the original statement

-l. 95-> NPC should be mentioned as an abbreviation of “non-parenchymal cells” the first time it appears in the text

Response: We stated at -1.50

-l.216-219 -> this sentence seems a little misplaced in the Result section, I suggest that the Authors move it in the Discussion

Response: We need to cite other previous research in ‘Results’ part to pave the way for our research content, so we prefer to maintain the original statement. However,based on reviewer’s kind suggestions, we simplified the description here.

-l.243 -> does “H/R” stand for ischemia/reperfusion? If that is the case, acronyms should be used consistently throughout the manuscript

Response: H/R means Hypoxia/reoxygenation, an in vitro model of I/R model.

-Figure 3E is too small, even zooming in I could not read the legend

Response: We zoomed in on the scale of 3E.

-l.279-281 -> unclear sentence

Response: Edited to read clearly

-l.340-343 ->  this sentence seems a little misplaced in the Result section, I suggest that the Authors move it in the Discussion, maybe rephrasing it with more technical language.

Response: Modified

-Figure 6B should have significance bars also comparing FL/FL and MyeKO

Response: Modified

We appreciate for Editors and Reviewers’ warm work earnestly, and hope that the correction will meet with approval.

Once again, thank you very much for your comments and suggestions.

Thank you and best regards.

Sincerely,

Liyong Pu

E-mail:

[email protected]

Round 2

Reviewer 1 Report

Authors answer all my queries